# Time-Gated Pulsed Raman Spectroscopy with NS Laser for Cultural Heritage

**Xueshi Bai** [1] and **Vincent Detalle** [1,2,*]

1 Centre de Recherche et de Restauration des Musées de France (C2RMF), 14 Quai François-Mitterrand, 75001 Paris, France
2 SATIE, Systèmes et Applications des Technologies de l'Information et de l'Energie, CY Cergy-Paris Université, Pôle SIAME, CNRS UMR 8029, 5 Mail Gay Lussac, 95031 Neuville sur Oise, France
* Correspondence: vincent.detalle@cyu.fr

**Abstract:** Raman spectroscopy, a non-destructive reference technique, is used in heritage science to directly identify materials like pigments, minerals, or binding media. However, depending on the material, the laser source can induce a strong fluorescence signal that may mask the Raman signal during spectral detection. This photo-induced effect can prevent the detection of a Raman peak. A pulsed Raman spectroscopy, using a time-gated detection and pulsed laser, is proven capable of rejecting the fluorescence background and working with the environmental light, which makes Raman spectroscopy more adapted for in situ applications. In this paper, we investigated how an ns pulsed laser can be an excitation source of Raman spectroscopy by focusing on different parameters of laser excitation and collection. With proper implementation, this pulsed Raman technique can be used for cultural heritage with an ns pulsed laser for the first time.

**Keywords:** time-gated Raman spectroscopy; pulsed Raman spectroscopy ns laser; heritage science; fluorescence rejection

## 1. Introduction

Raman spectroscopy is one of the most efficient nondestructive techniques for identifying materials and is applied in many different domains, especially for cultural heritage [1–10]. However, the laser source can generate a strong fluorescence when organic materials are present. Laser-induced fluorescence or, more generally, photoluminescence may mask the Raman signal during spectral detection, which means that no Raman peak can be determined by the spectra [11,12]. Therefore, in the 70s, the time-gated Raman spectroscopy technique was developed with a very short-pulsed laser to eliminate the impact of the fluorescence [13,14]. Harries et al. were the first team to compare the level of fluorescence background rejection in a time-gated experiment to that of continuous excitation on a fluorophore-doped benzene Raman band at 992 cm$^{-1}$ [14]. The ambient light does not interfere with time-gated Raman spectral results and improves the signal-to-noise ratio of weak Raman signals in the presence of fluorescence [14–17]. Its applications have emerged in different domains, such as mineralogy, planetary sciences, biotechnology, biomedical, pharmaceutical identification, hot temperature, environmental sensing, and process industrial sensing [13]. However, the pulsed Raman approach for in situ cultural heritage conservation applications has never been demonstrated in the literature.

Time-resolved Raman spectroscopy needs a photon detector with high sensitivity (single photon counting ability), a fast external trigger, and a temporal resolution in the sub-nanosecond range, such as intensified-charge-coupled device (ICCD) cameras, the most common detector for pulsed Raman. New, high-quality ICCD detectors can rapidly minimize the fluorescence contribution during laser-material interaction. This approach is important as conservation operation generally occurs during the day in normal ambient light, with huge possibility of light intensity or spectral variations. For example, an

intensified and gated detector allowed the complete removal of the environmental light by applying a gate time of 100 ns and a delay time of 12 ns [18]. The lifetime of the fluorophore is usually several nanoseconds, but an ICCD camera generally has a classical minimum acquisition time of 5 nm. Thus, implementing this technique is still challenging when using an ns laser. The detection window must be set wide enough to overlap as much of the excitation pulse as possible, maximizing the Raman contribution. However, it must also be set narrow enough to avoid significant fluorescence emission. A good compromise position should be to set the time window to overlap just the first half of the excitation pulse [12]. In addition, pulsed Raman spectroscopy is generally used at the micro-scale level for cultural heritage applications due to the mixed material technique for identifying pigments, binding media, and varnishes or dyes. However, its application in measurement in situ, like glass identification, allows it to work during normal daylight, which cannot be achieved by continuous Raman spectroscopy [6,9,19–21]. The signal-to-noise ratio is improved, and the ambient contribution light is removed, avoiding false light contribution.

This paper presents the different Raman measurements as a function of laser parameters and detection time windows. These experimental results show that the Raman signal can be improved compared to the fluorescence background by optimizing the parameters of laser excitation and acquisition. We also compare different laser wavelengths to find a good candidate for a multi-spectroscopic technique that can be applied to cultural heritage materials analysis.

## 2. Materials and Methods

The samples were chosen to represent many different materials commonly used in artworks. In this work, the tested samples varied between pure raw samples, such as Paraloid B72, and pigment lead white and raw materials, such as beeswax and white marble: each featured one organic and one inorganic. Paraloid B72 and beeswax are grain-shaped, the original commercial form. The pigment lead white is compressed into a powder pellet with 5-tons pressure for 1 min.

Concerning the different laser parameters, the pulse duration is one widely approximated by the community, considering commonly conventional descriptions.

As a conventional pulse parameter laser, a laser pulse's nominal duration is usually concerned with the full width at half maximum (FWHM). If the laser temple profile is considered a Gaussian function (Figure 1 dashed line):

$$f(x) = \exp\left(\alpha x^2 + \beta x + \gamma\right) \tag{1}$$

where $\alpha = -\frac{1}{2c^2}$, $\beta = \frac{b}{c^2}$, $\gamma = \ln a - \frac{b^2}{2c^2}$, a, b, and c are the constants.

The parameter $c$ is related to the FWHM of the peak according to

$$\text{FWHM} = 2\sqrt{2\ln 2}\,c \ \approx 2.35482c \tag{2}$$

We cite the parameter of full width at tenth of maximum (FWTM) for a Gaussian function as the duration of the laser presenting:

$$\text{FWTM} = 2\sqrt{2\ln 10}\,c \ \approx 4.29193c \tag{3}$$

Then, for a nominal duration of 7 ns, the experimental recording temporal profile of a 532 nm laser pulse is presented in Figure 1. The FWTM is about 13 ns, and the photons are measured for much longer than this duration, at least 20 ns, as measured in Figure 1 (solid line). The delay was set from 0 ns without any laser signal detected and a step of 2 ns. The ICCD detection window time was set at 4 ns, less than the minimum detection, which could achieve a few photons from the laser between opening and closing the multi-channel plates in front of the camera and also avoided damaging the camera. In this case, the time window was fewer than 1 ns.

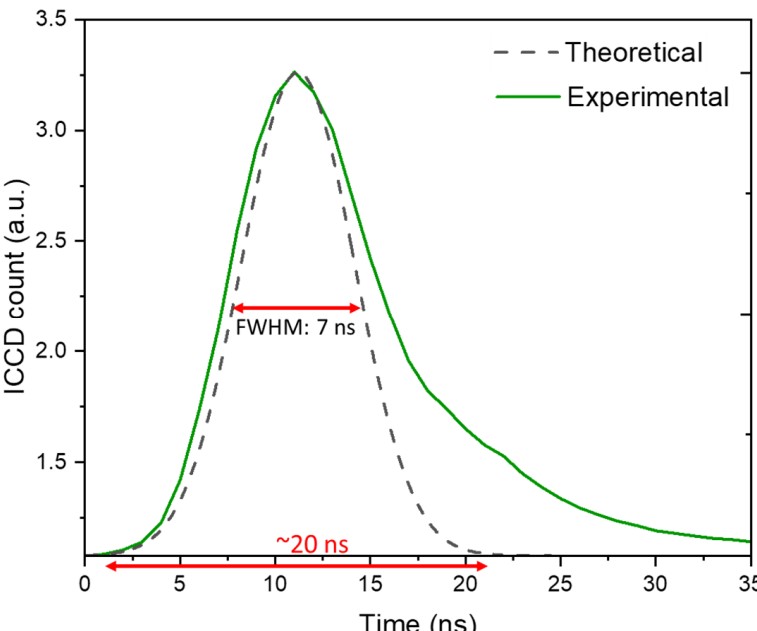

**Figure 1.** A Gaussian function describes the temporal profile of a 7 ns nominal duration of the laser pulse: theoretical curve (dashed line) and experimental curve (solid line).

The laser scattering process is quasi-instantaneous, in the picosecond order of time [15,17], so the Raman scattering occurs within several tens of nanoseconds and simultaneously with the laser pulse. Compared to the fluorescence with its typical lifetime of several nanoseconds or the phosphorescence between microseconds and milliseconds, if detection occurs only in the first part of the ns laser, fluorescence can be avoided. The lifetime of laser-induced fluorescence is in the tens of nanoseconds, so the fluorescence arrives later than the scattering light. We can take advantage of this time interval to reduce the fluorescence background in Raman spectra.

The continuous Raman system is a homemade micro-analysis system with a 532 nm laser diode, whose output laser power is about 1.4 mW, and a ×40 microscopic objective. The pulsed Raman was carried out with three laser wavelengths: 266 nm, 355 nm, and 532 nm from the harmonics of an Nd:YAG laser (Lumibird, Les Ulis, France), corresponding to a mean power of 4.25 mW, 1.14 mW, and 1.94 mW, respectively. The diameters of the pulsed laser were measured at about 200 μm.

The collection path consisted of an optical fiber placed in the image plane of the 4f imaging system for these two systems. This fiber was connected to a Czerny-Turner spectrometer (Shamrock 303i, Andor Technology, Oxford, UK) with three optical gratings of 600, 1200, and 1800 lines/mm. The spectrometer was equipped with an ICCD camera (DH340T-18F-E3, Andor Technology, Oxford, UK). The collection time window was set for the continuous Raman mode as 50 ms and for the pulsed Raman mode as 10 ns. Then, the pulsed Raman signal was accumulated from 2000 laser pulses, thus equivalent to 20 μs total exposition to the laser. The detection time window remained constant at 10 ns, except for the acquisition for comparison of detection time windows.

### 3. Results

*3.1. Raman Spectroscopy in Different Excitation Conditions*

3.1.1. Improvement of Raman Signal to Luminescence Background by Pulsed Laser

Subtraction of the ambient light spectrum from the Raman spectrum with the ambient light is now generally used when the ambient light cannot be turned off during Raman measurement. Our previous work showed that if the ambient light is stable, this subtraction operation may make the Raman spectrum appear [21]. However, when the ambient light varies, which is often the case for in situ measurement, a pulsed laser can carry out Raman

spectroscopy measurement without ambient light infection. Compared with the pulsed laser excitation in Figure 2a, the Raman signal can also give a better definition of the peaks, such as 964, 1024 and 1110 cm$^{-1}$.

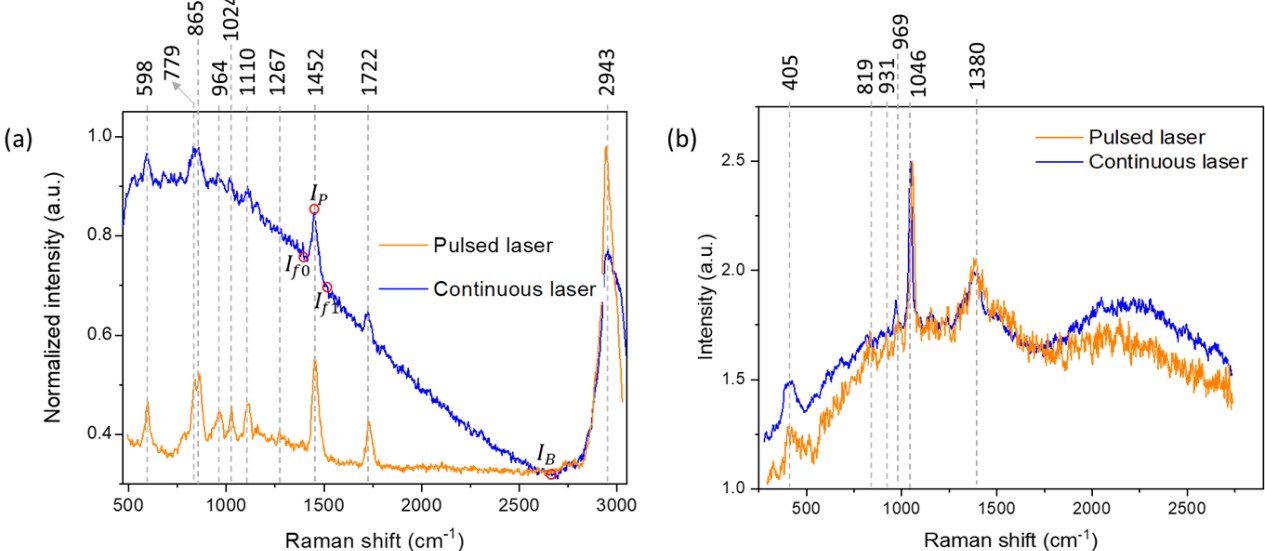

**Figure 2.** Comparison of Raman spectra from (**a**) the Paraloid B72 and (**b**) lead white sample in continuous and pulsed laser excitation. $I_P$: Intensity of Raman peak at 1450 cm$^{-1}$ (C-H bend), $I_{f0}$ (1366 cm$^{-1}$), and $I_{f1}$ (1530 cm$^{-1}$): Intensity of fluorescence adjacent to the Raman peak, $I_B$ : Intensity of background at 2660 cm$^{-1}$ where no fluorescence emits.

To evaluate the improvement of the Raman to fluorescence ratio (RFR), we define a parameter as

$$\text{RFR} = \frac{I_P - I_F}{I_F - I_B},\tag{4}$$

where $I_P$: intensity of Raman peak; $I_F = \left( I_{f0} + I_{f1} \right)/2$: intensity of fluorescence at the position of the Raman peak at 1452 cm$^{-1}$ (C-H bend), the mean value calculated by the intensity at position $f_0$ and $f_1$ at 1366 cm$^{-1}$ and 1530 cm$^{-1}$, respectively; $I_B$: intensity of background at 2660 cm$^{-1}$, where there is neither fluorescence emission nor Raman spectroscopy. Therefore, for the continuous Raman, the RFR≈0.23, and the pulsed Raman, RFR ≈0.80 at the Raman peak of 1452 cm$^{-1}$. This means an improvement of about four times versus the Raman to fluorescence ratio. Here, the mentioned fluorescence contains the fluorescence and also phosphorous light for the continuous Raman mode. The phosphorous light can be eliminated automatically for several nanoseconds' detection time windows because its lifetime is about µs even until ms. With time-gated detection, the most intense fluorescence has been removed from the signal of Raman spectroscopy since it arrives later, outside the detection duration.

In another case, for the materials that do not have strong luminescence in the Raman spectral zone, such as lead white, the pulsed Raman gives a similar Raman signal as the continuous one, shown in Figure 2b. The RFR is calculated for the Raman peak at 969 cm$^{-1}$. In Formula (4), the $I_B$ is taken as 0 since the background cannot be defined. For the continuous Raman, the RFR ≈ 0.07, and the pulsed Raman, RFR ≈ 0.08. Therefore, the pulsed laser improves the Raman signal slightly in this case. When the sample emits no luminescence in the Raman spectral range, the pulsed Raman can still avoid pollution from the ambient light, so the signal is improved in this term.

3.1.2. Impact of Laser Wavelength and Intensity on Pulsed Raman Spectroscopy

When "Raman spectroscopy" refers to vibrational Raman using laser wavelengths which are not or hardly absorbed by the sample, the intensity of Raman scattering is defined by the following equation:

$$I = Kl\alpha^2 \, \omega^4 \ \sim \ \left(\frac{1}{\lambda^4}\right), \tag{5}$$

where $K$ consists of constants such as the speed of light, $l$ is the laser power, $\omega$ is the frequency of the incident radiation, and $\alpha$ is the polarizability of the electrons in the molecules [22]. $\lambda$ is the laser wavelength. The strength of the Raman signal increases with the fourth power of the frequency $\omega \ \sim 1/\lambda$ of the incident light. In addition, Raman scattering is an inelastic process involving only one in $10^6$–$10^8$ of the photons scattered among the incident ones. [22,23] Photons with shorter wavelengths are more energetic. When they lose the same energy after inelastic scattering as those of longer wavelengths, the difference in wavelength between the incident light and the scattered ones becomes much smaller than the laser with a longer wavelength. Therefore, a high-resolution spectrometer is needed to precisely determine the Raman peak. Moreover, a special detector, rather than a classical CCD camera, is required when using an infrared laser. Thus, the adaptation in terms of producing these spectral instrumentations is more demanding today.

Therefore, when an ns pulsed laser is used to perform a Raman spectroscopy, laser wavelength and intensity can directly involve Raman spectra. Moreover, the laser pulse with higher energy can give greater Raman scattering signal, but the fluorescence can also be more potent, even strong enough to overwhelm the Raman signal. According to Formula (5), the shorter laser wavelength is more attractive for the scattering process, but it also induces more fluorescence that may mask the Raman signal during spectral detection.

In the Paraloid B72 sample, shown in Figure 3, the Raman peaks were well observed under 355 nm and 532 nm with optical grating of 1800 lines/mm and 600 lines/mm, respectively. To identify the whole range of Raman spectra from 355 nm excitation, the spectrum consists of three different measured positions of the grating diffraction. The 266 nm laser requires the spectrometer to have an even better resolution than that given by an optical grating of 1800 lines/mm, which was not provided in our laboratory. The 266 nm pulsed Raman spectra were therefore not observable, although more efficient for inelastic scattering.

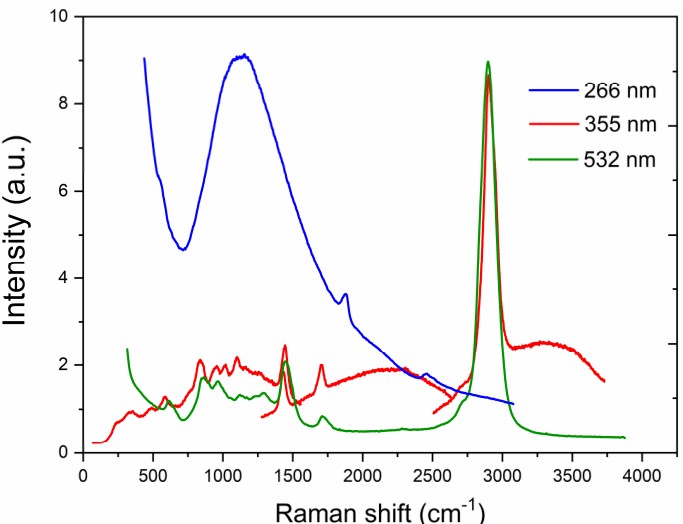

**Figure 3.** Raman spectra of Paraloid B72 obtained by three wavelengths of Nd:YAG laser harmonics: 266, 355, and 532 nm.

If the fluorescence can be removed correctly, Figure 4 shows the pulsed Raman spectra obtained by a 355 nm and 532 nm laser on the Paraloid 72 sample (Figure 4a) after subtraction of the spectra fluorescence baseline (Figure 3) and white marble (Figure 4b) from

the previous work [21]. The Raman peaks were well observed by these two laser wavelengths; it seems the fluorescence had little impact with this experimental configuration and time-gated detection.

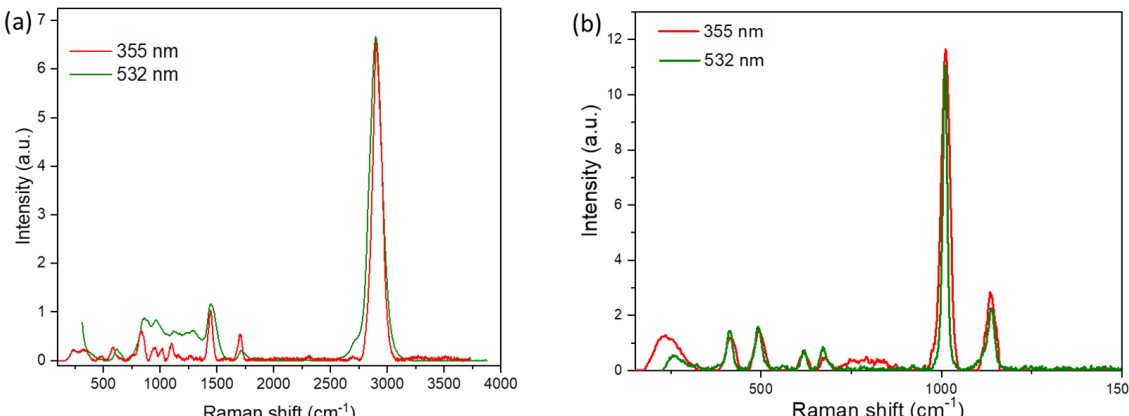

**Figure 4.** The pulsed Raman spectra obtained by a 355 nm and 532 nm laser on the sample (**a**) Paraloid 72 after subtraction of the spectra fluorescence baseline in Figure 3 and (**b**) white marble [21].

The 355 nm pulsed laser wavelength seems a good compromise for performing the pulsed Raman analysis because it excites materials that will fluoresce in the spectral range when separated from the Raman scattering emission. However, the beginning of this fluorescence may also impact the quality of Raman spectra. To avoid the fluorescence as much as possible, a longer wavelength than 532 nm is needed, such as the fundamental emission of Nd:YAG at 1064 nm. However, unlike the 266 nm that needs a high-resolution spectrometer, using a near-IR laser, we require a detector more efficient in the IR range instead of a CCD.

### 3.1.3. Impact of Pulsed Laser Energy (Power) on Raman Spectroscopy

According to Formula (5), laser intensity also plays an essential role in Raman scattering. The detection time window was fixed at 5 ns instead of 10 ns from 2 ns after the arrival of the laser pulse, to eliminate the fluorescence as much as possible. Figure 5 shows the Raman spectra from beeswax with different laser pulse energy at 532 nm. Higher laser power can help to increase the signal-to-fluorescence ratio and the very early time window is also required to limit the fluorescence emission.

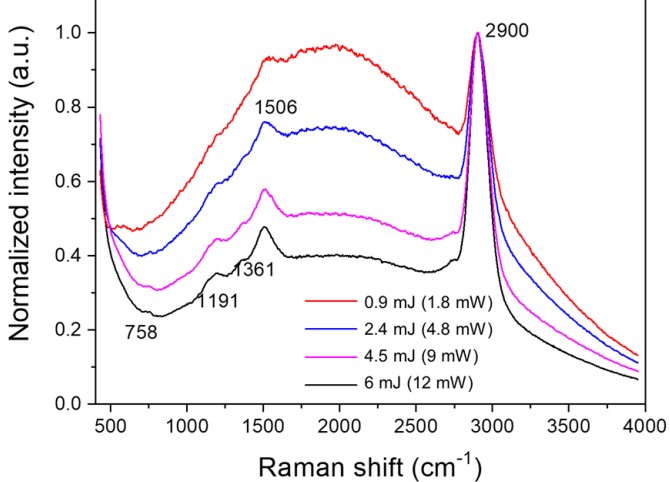

**Figure 5.** Raman spectra from beeswax with different laser pulse energy (power) and a detection window of 5 ns.

However, for fragile materials, it cannot exceed the damage threshold for the Raman measurement. Here, a Gaussian laser temporal profile was used; if a laser pulse varies its energy during the window in another way, the irradiance should be considered instead of averaged power. In addition, the spatial profile of the laser beam needs to be considered, to avoid the hot point inside the laser beam damaging the materials.

### 3.2. Effect of Detection Window Selected for Time-Resolved Raman Spectroscopy

The fluorescence emission evaluates as a function of time and the Raman spectroscopy because the laser pulse irradiance varies according to the delay. Therefore, the detection time window for Raman spectroscopy with an ns laser pulse should be optimized according to the fluorescence lifetime and the Raman scattering feature.

Raman spectra from Paraloid B72 with a high resolution induced by a 532 nm laser pulse recorded by different delays are illustrated in Figure 6. The detection windows were fixed at 10 ns. From delay −10 ns until 10 ns, the fluorescence center shifted to the red part, giving a good Raman signal. The strongest signal to fluorescence occurs at about −2 or 0 ns of delay after the laser pulse arrived on the sample that recorded between 0 and 8 ns or between 0 and 10 ns Raman spectra, which corresponds to the very first half of the laser pulse.

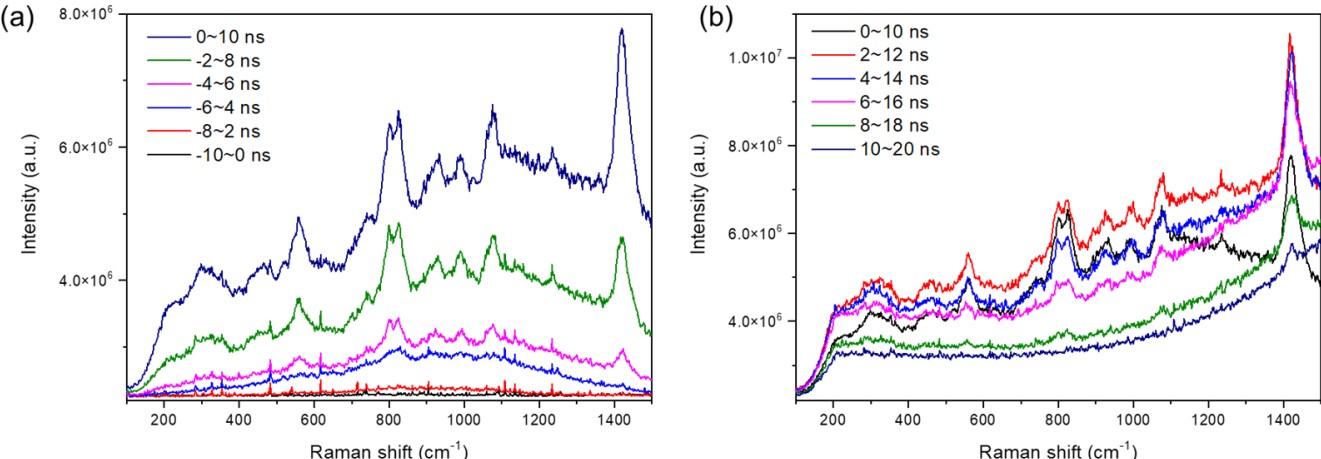

**Figure 6.** Raman spectra from Paraloid B72 with a high resolution induced by 532 nm laser pulse recorded by different delays (**a**) −10 ns to 0 ns and (**b**) 0 ns to 10 ns. The detection windows were fixed at 10 ns, and the minus value means the detection started before the laser pulse arriving the sample surface.

Global spectra yield the fluorescence increasing, with the Raman signal increasing as well, and the center shifting allows the detection of Raman scattering instead of covering the signals, especially with an efficient UV laser to generate the fluorescence. The strongest fluorescence emission situates out of the Raman spectral range (Figure 7), making it possible to acquire the time-resolved Raman and fluorescence spectroscopy simultaneously.

These two ns lasers, corresponding to the temporal profile shown in Figure 1, do not contain sufficient photons for an observable Raman scattering during the first 5 ns of the laser pulse with our current equipment. As the intensity increases during the laser pulse, the Raman peaks become visible and have a good signal-to-background ratio. At the same time, the fluorescence passes over the lifetime and becomes stronger and stronger. So, according to the analyzed materials, recording the Raman signal must be done before the fluorescence emission reaches its maximum.

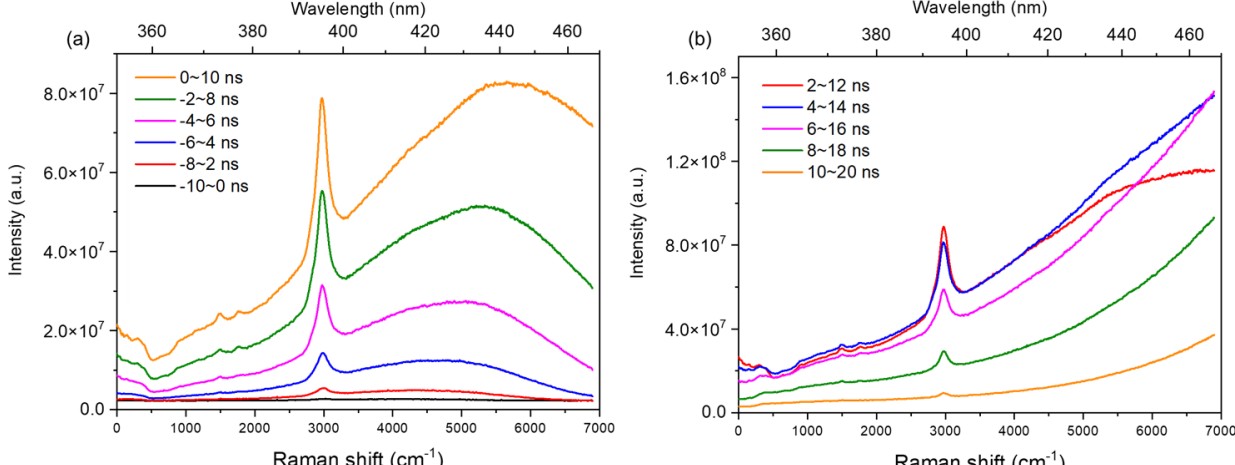

**Figure 7.** Global Raman spectra from Paraloid B72 with fluorescence emission induced by 355 nm (23 mJ/cm$^{-2}$) laser pulse recorded by different delays (**a**) −10 ns to 0 ns and (**b**) 2 ns to 10 ns. The detection windows were fixed at 10 ns, and the minus value means the detection started before the laser pulse arriving the sample surface.

## 4. Discussions

Raman spectroscopy carried out by a pulsed laser of several nanoseconds' duration has yet to be greatly studied in literature. The lifetime of induced fluorescence is usually in the order of time duration, making Raman measurement difficult, in addition to the diversity of cultural heritage object materials. Pulsed Raman spectroscopy can reject the total phosphorescence and environmental light. It also reduces the induced fluorescence by just taking the first part of the laser pulse of its temporal profile.

With the help of the ICCD camera, the detection time window can be fixed before the strong fluorescence accruing. The pulsed Raman showed the fluorescence was minimized on the inorganic and organic materials from natural and synthetic products, and the Raman-to-fluorescence ratio increased about four times.

From the results, the detection time window should be set before the strong fluorescence starts, so for an ns laser with FWHM of 7 nm at 532 nm, the optimized time window is between 2 and 10 ns after the laser arrives on the surface. Laser parameters are the decisive parameters for the Raman scattering; higher energy can induce stronger fluorescence, but if the detection time window is well selected or uses a laser wavelength that induces fluorescence out of the Raman peaks range, a laser pulse with more intensity can give a better Raman signal. Dealing with high fluorescence materials, the fundamental emission of Nd:YAG requires a detector in the infrared spectral range. As shown in the results of laser energy effects, the more intense laser pulse may help to obtain better Raman scattering. Still, for cultural heritage materials that are usually sensitive and fragile, the second and third harmonic of an Nd:YAG laser may lead to damage [24]. The development of a new detector with good efficiency in the infrared range should be used in cultural heritage. The shorter laser wavelength is more efficient for Raman scattering, but also needs a high-resolution spectrometer.

When laser energy is well controlled, the spectroscopic technique can switch between them: the low energy of a UV pulsed laser efficiently excites the material fluorescing, then more energy may give a better Raman signal without material damage. The energy continues to increase and the laser-induced breakdown spectroscopy (LIBS) can be carried out to elemental analysis. This combination allows elemental and molecular analysis at the same point on cultural heritage objects [19,21].

## 5. Conclusions

In this paper, we demonstrated that an ns laser with short wavelengths, such as 355 nm and 532 nm, is suitable to perform pulsed Raman spectroscopy analysis under ambient light, which cannot be reached with a continuous laser. As a compromise, the 355-nm laser pulse has been proven suitable to perform Raman analysis. This wavelength is commonly used to induce fluorescence in the visible spectral range. Therefore, such a UV laser can be used in the device, coupling different spectroscopic techniques, for mobility and multi-analytical ability at the same point. This implementation can be applied to cultural heritage and other domains that need to perform Raman analysis under ambient light.

**Author Contributions:** Methodology, software, validation, formal analysis, investigation, resources, data curation, writing—original draft preparation, writing—review and editing, and visualization: X.B. and V.D.; Conceptualization, supervision, project administration, and funding acquisition: V.D. All authors have read and agreed to the published version of the manuscript.

**Funding:** This research was funded by the French Ministry Research Program EquipEx PATRIMEX (ANR-EQPX-0034) and ESR ESPADON-PATRIMEX (ANR-21-ESRE-00050) and funded as IPERION-CH project by the European Commission, H2020- INFRAIA-2014-2015, Grant No. 654028 and IPERION-HS project by the European Commission, H2020- INFRAIA-2019-1, Grant No. 871034. The research work had the support of a grant under the Decree of the Government of the Russian Federation No. 220 of 9 April 2010 (Agreement No. 075-15-2021-593 of 1 June 2021).

**Data Availability Statement:** The data are available upon reasonable request.

**Conflicts of Interest:** The authors declare no conflict of interest.

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
