# Peer review of "Time-Gated Pulsed Raman Spectroscopy with NS Laser for Cultural Heritage"

_heritage, doi:10.3390/heritage6020082_

Round 1

Reviewer 1 Report

The manuscript "heritage-2168039" may be published after revision. Some suggestions of improvement:

1. At Materials and methods section keep subsection 2.2 and move subsections 2.1.1. and 2.1.2., or part of them, either in the Introduction or Results and discussion or both. 

2. Indicate how figure 1 was obtained or give references.

3. line 98: "a" should be "alpha"

4. When using Raman spectroscopy it is custom to give laser power. Please give the laser power for the measurements.

5. Revise English, especially topic and typos. The phrase at lines 18-21 is too long and incomplete.

Author Response

Dear reviewer,

Thank you for helping us improve this article with your comments and suggestions. We answered point to point as follows, you can find also the response in the attachment file in red police.

The manuscript "heritage-2168039" may be published after revision. Some suggestions of improvement:

  1. At Materials and methods section keep subsection 2.2 and move subsections 2.1.1. and 2.1.2., or part of them, either in the Introduction or Results and discussion or both. 

We have merged subsection 2.1.2 in the Results and left subsection 2.1.1 in order to explain the time detection parameters chosen in this section “method”.

  1. Indicate how figure 1 was obtained or give references.

We added the text to explain how we measured the pulse evolution: “Then, for a nominal duration of 7 ns, the experimental recording temporal profile of a 532 nm laser pulse is presented in Figure 1. The FWTM is about 13 ns, and the photons are measured much longer than this duration, about at least 20 ns measured in Figure 1 (solid line). The delay was set from 0 ns without any laser signal detected with a step of 2 ns. The ICCD detection window time was set at 4 ns, less than the minimum detection, which can get a few photons from the laser between opening and closing the multi-channels plates in front of the camera and also avoid damaging the camera, so in this case, the time window of fewer than 1 ns.”

  1. line 98: "a" should be "alpha"

It is corrected

  1. When using Raman spectroscopy it is custom to give laser power. Please give the laser power for the measurements.

For a pulsed laser, the mean power can be calculated by pulse energy and repetition rate. So we add the mean power in the text, as “The pulsed Raman was carried out with three laser wavelengths: 266 nm, 355 nm, and 532 nm from the harmonics of Nd: YAG laser, corresponding to a mean power of 4.25 mW, 1.14 mW and 1.94 mW, respectively”.

  1. Revise English, especially topic and typos. The phrase at lines 18-21 is too long and incomplete.

We have revised English.

Sincerely

Reviewer 2 Report

In this manuscript titled “Time-gated Raman spectroscopy with ns laser pulses for cultural heritage”, X. Bai and V. Detalle, described Time-gated Raman spectroscopy with ns laser pulses for potential applications in cultural heritage. In my opinion this paper does not add any significant knowledge due to several flaws and lack of novelty in the manuscript. Therefore, major revision is required as follows before the manuscript can be accepted.

1.      Introduction section should be improved and novelty statement must include in the introduction section including comprehensive literature review

2.      Why Paraloid B72 and pigment lead white samples were used in this work? Any specific reason to choose these samples?

3.      In Fig. 2 (a), the scale on y-axis should kept fixed for better comparison.

4.       In Fig. 2, 3 and 4, the Raman peaks should be labelled on the spectra.

5.      On page. 5, line 171-172, it in mentioned that “For the continuous Raman, the RFR = 0.07, and the pulsed Raman, RFR=0.08. So, the pulsed laser can improve the Raman signal slightly in this case”. It can be seen from RFR that no significant difference is noticed then how the author claimed that pulse laser improve Raman signal relative to continuous Raman?

6.      In section 3.2, the detection window was fixed at 10 ns. Any particular reason for that?

7.      I would suggest the authors to summarize only the findings in the conclusion section and the rest explanation should include in separate discussion section.

8.      The English in whole manuscript should be improved, for example, sentence 188-191 and 201-205 are difficult to understand.

Author Response

Dear reviewer,

Thank you for helping us improve this article with your comments and suggestions. We answered point to point as follows, you can find also the response in the attachment file in red police.

In this manuscript titled “Time-gated Raman spectroscopy with ns laser pulses for cultural heritage”, X. Bai and V. Detalle, described Time-gated Raman spectroscopy with ns laser pulses for potential applications in cultural heritage. In my opinion this paper does not add any significant knowledge due to several flaws and lack of novelty in the manuscript. Therefore, major revision is required as follows before the manuscript can be accepted.

  1. Introduction section should be improved and novelty statement must include in the introduction section including comprehensive literature review

We have checked the Introduction section. In this section, we cited the main fundamental and application articles of time-gated Raman spectroscopy. We rewrite the English of this part to make it better understood. And we added the text “But the use of the pulse Raman approach for in situ for cultural heritage conservation applications has never been demonstrated in the literature….. The capability of new high-quality ICCD detectors is well controlled for minimizing the fluorescence contribution at a short time of the laser material interaction. The importance of this approach is due to the fact that conservation operation generally occurs during the day in normal ambient light with a huge possibility of light intensity or spectral variations…. Moreover, its capability to allow characterizing stained glaze during its emerging with this approach. It is not only the signal-to-noise ratio that is improved but also the ambient contribution light that is removed avoiding false light contribution.” in the Introduction as general remarks about the novelty of this article.

  1. Why Paraloid B72 and pigment lead white samples were used in this work? Any specific reason to choose these samples?

We wanted to choose samples representing organic and inorganic materials. Paraloid B72 is used as a conservation product for more than 20 years as varnish or fixative/consolidating products for a mural or easel painting applications. Moreover, it can be used on stone or render consolidating products as well. White lead is the basic lead carbonate 2PbCO3·Pb(OH). It is a complex salt, containing both carbonate and hydroxide ions. White lead occurs naturally as a mineral, in which context it is known as hydrocerussite, a hydrate of cerussite. It was formerly used as an ingredient for the painting layer or for the preparation layer in mural paintings or easel paintings and a cosmetic called Venetian ceruse, because of its opacity and the satiny smooth mixture it made with dryable oils since antiquity.

For these reasons, in this paper, we used these both products as good candidates for representative artwork conservation material.

  1. In Fig. 2 (a), the scale on y-axis should kept fixed for better comparison.

The absolute ICCD counts cannot compare the different intensities of Raman spectra between the continuous and pulsed laser, because the numbers of initial photons are not the same. So, in order to compare these two configurations, the normalized intensity by their own maximum is used to replace the original figure in the manuscript.

  1. In Fig. 2, 3 and 4, the Raman peaks should be labelled on the spectra.

We labeled the Raman peaks in Figure 2. Figures 3 and 4 are the results from the same samples and the same peaks, so we leave the spectra clear in order to easier to compare.

  1. On page. 5, line 171-172, it in mentioned that “For the continuous Raman, the RFR = 0.07, and the pulsed Raman, RFR=0.08. So, the pulsed laser can improve the Raman signal slightly in this case”. It can be seen from RFR that no significant difference is noticed then how the author claimed that pulse laser improve Raman signal relative to continuous Raman?

The improvement is significant when the laser-induced luminescence is strong. When the sample emits no luminescence in the Raman spectral range, the pulsed Raman can avoid pollution from the ambient light, so the signal is improved in this term.

We added “So, the pulsed laser improves the Raman signal slightly in this case. When the sample emits no luminescence in the Raman spectral range, the pulsed Raman can still avoid pollution from the ambient light, so the signal is improved in this term.” in the text.

  1. In section 3.2, the detection window was fixed at 10 ns. Any particular reason for that?

The smallest time window allowed by the ICCD camera itself is 5 ns. In order to minimize the readout noise and also to have sufficient Raman signal, the detection window was chosen as 10 ns.

  1. I would suggest the authors to summarize only the findings in the conclusion section and the rest explanation should include in separate discussion section.

We followed the suggestion by separating the discussion and conclusion sections

  1. The English in whole manuscript should be improved, for example, sentence 188-191 and 201-205 are difficult to understand.

We have checked the English in the whole manuscript.

Sincerely

Round 2

Reviewer 2 Report

The author addressed answers of all questions in a satisfactory manner.